# Gingival Crevicular Fluid Zinc- and Aspartyl-Binding Protease Profile of Individuals with Moderate/Severe Atopic Dermatitis

**DOI:** 10.3390/biom10121600

**Published:** 2020-11-26

**Authors:** Fernando Valenzuela, Javier Fernández, Marcela Aroca, Constanza Jiménez, Daniela Albers, Marcela Hernández, Alejandra Fernández

**Affiliations:** 1Department of Dermatology, Faculty of Medicine, University of Chile, Santiago 8380453, Chile; fernidando@u.uchile.cl; 2Centro Internacional de Estudios Clínicos, Probity Medical Research, Santiago 8420383, Chile; 3Dermatology Unit, San José Hospital, Santiago 8380419, Chile; javi.fernandez@uc.cl; 4Department of Oral Pathology, Faculty of Dentistry, Universidad Andres Bello, Santiago 8370133, Chile; M.aroca@uandresbello.edu (M.A.); c.jimenezlizama@uandresbello.edu (C.J.); 5Department of Statistics, Faculty of Dentistry, Universidad Mayor, Santiago 8320000, Chile; Daniela.albers@mayor.cl; 6Laboratory of Periodontal Biology, Faculty of Dentistry, Universidad de Chile, Santiago 8380544, Chile; mhernandezrios@odontologia.uchile.cl; 7Department of Oral Pathology and Medicine Faculty of Dentistry, Universidad de Chile, Santiago 8380544, Chile

**Keywords:** enzymes, peptide hydrolases, metalloproteases, biomarkers, gingival crevicular fluid, atopic dermatitis

## Abstract

Atopic dermatitis (AD) is a protease-modulated chronic disorder with heterogenous clinical manifestations which may lead to an imprecise diagnosis. To date, there are no diagnostic protease tests for AD. We explored the gingival crevicular fluid (GCF) protease profile of individuals with moderate/severe AD compared to healthy controls. An exploratory case-control study was conducted. AD patients (*n* = 23) and controls (*n* = 21) were enrolled at the International Center for Clinical Studies, Santiago, Chile. Complete dermatological and periodontal evaluations (involving the collection of GCF samples) were made. The levels of 35 proteases were analyzed using a human protease antibody array in matching AD patients (*n* = 6) and controls (*n* = 6) with healthy periodontium. The GCF levels of zinc-binding ADAM8, ADAM9, MMP8, Neprilysin/CD10, aspartyl-binding Cathepsin E, serin-binding Protein convertase9, and uPA/Urokinase proteases were lower in moderate/severe AD patients compared to controls (*p* < 0.05). No inter-group differences in the levels of the other 28 proteases were found. MMP8, Cathepsin E, and ADAM9 were the biomarkers with the highest sensitivity and specificity regarding the detection of AD (*p* < 0.05). The area under receiver operating characteristic (ROC) curve for MMP8 was 0.83 and MMP8 + ADAMP9 was 0.90, with no significant differences (*p* = 0.132). A combined model of MMP8, Cathepsin E, and ADAM9 was not considered since it did not converge. Then, levels of MMP8 in GCF were determined using a multiplex bead immunoassay in 23 subjects with AD and 21 healthy subjects. Lower levels of MMP8 in the GCF from the AD group versus healthy group (*p* = 0.029) were found. This difference remained significant after adjustment by periodontitis (*p* = 0.042). MMP8 revealed the diagnostic potential to identify AD patients versus healthy controls, (ROC area = 0.672, *p* < 0.05). In conclusion, differences in the protease profile between AD and control patients were associated with MMP8, Cathepsin E, and ADAM9. Based on the multiplex assay results, MMP8 was lower in AD patients than controls, suggesting that MMP8 may be a diagnostic biomarker candidate.

## 1. Introduction

Atopic dermatitis (AD) is the most common chronic inflammatory skin disease in humans worldwide [1]. The prevalence of AD in young adults is high, with an estimated value between 10% and 34.1%, and its persistence over time is common, greatly affecting the quality of life of those who suffer from it [2].

The etiology of AD is multifactorial and involves the interaction of environmental, genetic, and immune system factors. The pathogenesis of AD is complex and combines cutaneous barrier dysfunction, systemic immune dysregulation, dysbiosis of the bacterial microbiome of the skin, and genetic factors. Initially, atopic dermatitis patients have a predisposition to maintain the T helper (Th) 2 lymphocyte response, while a change in the response from Th2 to Th1 promotes the chronicity of the disease [1,3].

In addition to the immune-inflammatory response, the abnormal expression or activity of proteases has been associated with the pathogenesis of AD. Many zinc- and serine-binding proteases are upregulated or downregulated in AD lesion/serum samples and seem to play a role in the disruption of the normal skin-barrier function. Moreover, the increased protease activity in AD leads to permeability barrier dysfunction, inflammation, and defects in the antimicrobial barrier. Due to the proteolytic effects, external antigens of AD can easily invade the epidermis, resulting in dermatitis, coupled with the induction of Th2 cytokines [4,5,6].

AD clinical manifestations are heterogeneous, and its clinical diagnosis is highly complex. Severely pruritus is the most frequent clinical manifestation and is triggered by heat, perspiration, emotional stress, etc. [3]. The clinical characterization of AD individuals might not satisfactorily reflect the pathophysiologic profile of patients with AD. The use of biomarkers may help in better defining the clinical heterogeneity of the disease and contribute to its treatment. Currently, there are no biomarkers that can differentiate the disease from other entities or indicate the clinical status of AD in adults [3,4,5,6,7].

The gingival crevicular fluid (GCF) is a transudate harvesting component from local periodontal tissues and serum that can be obtained from the gingival crevice surrounding the teeth. GCF represents a source of a wide range of biological molecular markers allowing the diagnosis, monitoring, prognosis, and severity of several diseases in a non-invasive approach [8]. Accordingly, a recent study supports the feasibility of measuring endothelial and placental biomarkers in maternal GCF [9]. Moreover, biomarkers of inflammation have been considered as a biomarker in the progression of coronary heart disease and chronic periodontitis [10]. Therefore, we hypothesize that atopic dermatitis patients show a differential GCF protein profile from healthy individuals, allowing the identification of potential biomarkers for point-of-care monitoring of atopic dermatitis in adults. Hence, the aim of this study was to explore the protease profile of GCF samples in individuals with moderate/severe atopic dermatitis in relation to healthy controls.

## 2. Materials and Methods

### 2.1. Study Design

This case control study was approved by the Bioethics Committee of the Faculty of Dentistry of Universidad Andres Bello (no. PROPRGFO_002001980). All individuals were informed about the objective of the investigation and signed an informed consent. The whole process was in accordance with the ethical standards of the institutional and/or national research committee and with the 1975 Helsinki declaration, and its later amendments or comparable ethical standards.

### 2.2. Population and Dermatological Evaluation

Patients with atopic dermatitis and systemically healthy controls evaluated at the International Center for Clinical Studies (CIEC) Santiago, Chile, between March and July of 2019 were selected for convenience. The diagnosis of atopic dermatitis was based on the medical history and physical examination made by a dermatologist. Measured clinical atopic dermatitis variables included: The Scoring Atopic Dermatitis (SCORAD) scale, the Eczema Area and Severity Index (EASI), the Body Surface Area (BSA) index, and the Investigator Global Assessment scale for Atopic Dermatitis (IGA). Inclusion criteria were adults with or without a clinical diagnosis of AD and otherwise healthy. The exclusion criteria were smokers, pregnant women, patients with any concomitant skin pathology, such as Psoriasis, Seborrheic Dermatitis, Contact Dermatitis, or individuals who had been treated with antibiotics, non-steroidal inflammatory drugs, immunomodulators orally or intravenously within the previous 3 months. The included patients were matched by gender and age.

### 2.3. Periodontal Evaluation

Intra-oral clinical examinations were conducted by a periodontics specialist at the clinic. The periodontal examination was made with a manual periodontal probe. Clinical in full-mouth recordings at six sites in all teeth, including probing depths, the clinical attachment level, and bleeding on probing at the base of the crevice, excluding third molars. The periodontal disease was defined in accordance to the classification devised by Eke et al. [11]. Periodontally-healthy subjects were defined based on the absence of neither mild, nor moderate or severe periodontitis. Mild periodontitis was diagnosed when ≥2 interproximal sites had clinical attachment levels ≥ 3 mm and ≥2 interproximal sites had probing depths ≥ 4 mm (in a different tooth). Moderate periodontitis was defined when ≥2 interproximal sites had clinical attachment levels ≥ 4 mm (not on the same tooth) or/and ≥2 interproximal sites had probing depths > 5 mm. Finally, severe periodontitis was defined as ≥2 interproximal sites with clinical attachment levels > 6 mm (not on the same tooth) and ≥1 interproximal sites with PD > 5 mm.

### 2.4. Sample Collection

GCF samples were collected by a qualified periodontist using sterile periodontal strips (Periopaper^®^, Interstate Drug Exchange, Amityville, NY, USA). Teeth were isolated with cotton rolls and then, carefully dried with an air-syringe to prevent saliva contamination. Samples were collected by inserting the periodontal strips into the gingival sulcus of the first molar of each quadrant for 30 s. The four strips obtained per each individual were pooled. Afterwards, GCF samples were stored at −80 °C until analysis.

### 2.5. Elution of Proteins from Gingival Crevicular Fluid

Proteins from the GCF were extracted using a protein elution buffer (50 mM Tris-HCl pH 7.5, 0.2 mM NaCl, 5 mM CaCl_2_, and 0.01% Triton X-100) prepared into sterile tubes. Samples were incubated for half an hour at 4 °C and then centrifuged at 1.2 × 10^4^× *g* for 5 min at 4 °C. The procedure was repeated twice, and then the samples were frozen and kept at −20 °C until analysis, to reach a final elution volume of 160 µL.

### 2.6. Protease Antibody Array

An amount of 50 μL of aliquots were mixed with a cocktail of biotinylated detection antibodies and incubated overnight with the Proteome Profiler Human Protease Array Kit (R&D Systems, Minneapolis, MN, USA). The nitrocellulose membranes were washed to remove the unbound material. Streptavidin-HRP and chemiluminescent detection reagents were applied, and a signal was produced at each capture spot corresponding to the amount of protein bound. The membranes were exposed to an X-ray film for 5 min and the signal intensities were quantified using a densitometric software (Bruker MI SE, Billerica, MA, USA). The signal was measured in pixel density of the pair of duplicate spots representing each protease. A clear area of the array was used as a negative control. The relative expression level (REL) of the proteases was calculated using the following formula: REL = (cytokine signal intensity − mean intensity of negative control/mean intensity of positive control − mean intensity of negative control) × 100 [12].

### 2.7. Multiplex Bead Immunoassay

All GCF samples were diluted (2:50) with the assay buffer provided by the manufacturer in the multiplex bead immunoassay panel kit (Human Magnetic Luminex Assay^®^, R&D Systems, Minneapolis, MN, USA). Concentrations of MMP8 were measured in 96-well solid plates according to the manufacturers’ instructions. Data from the multiplex analysis panel were read using a digital platform (Magpix, Millipore, St. Charles, MO, USA) and then, analyzed using the MILLIPLEX AnalystR software (v5.1, Viagene Tech, Carlisle, MA, USA).

### 2.8. Sample Size

Due to the exploratory nature of this study, and being the first study of this kind to explore the GCF protease profile in patients with AD, we begun by screening the GCF profile in six periodontally-healthy AD patients and six periodontally-healthy controls matching in age, gender, and tobacco use. Periodontally affected subjects were excluded from this part of the study due to the effect that periodontitis and periodontal inflammation has on the concentrations of several biological markers in the GCF [9]. Then, from the results obtained, we proceeded to calculate the sample size needed to corroborate these results by means of a bead-based immunoassay. This method is a sensitive and highly specific method of evaluation for our analytes and has been widely validated in the literature for exploration of biomarkers in clinical diagnostics. From the results of the MMP8 protease profile, the sample size was calculated with an effect size of 1.1, with a level of significance of 0.05, and power of 0.8 determining a minimum sample size of 15 subjects with atopic dermatitis and 15 healthy controls.

### 2.9. Statistical Analysis

For protease antibody array results, the non-parametrical analysis was performed with the Mann-Whitney U test. The biomarker levels were dichotomized using the median as a threshold to determine the diagnostic ability and Odds ratios of each biomarker. The performance discrimination and diagnostic accuracy were evaluated through the construction of a receiver operating characteristic (ROC) curve, by calculating the area under the curve (AUC) of AD patients versus controls after logistic regression modeling.

For multiplex bead immunoassay results, MMP8 concentrations were transformed to log_e_, fulfilling normality, and homoscedasticity assumptions. The analysis was performed using the Fisher’s exact test, *t*-test, and a multiple linear regression model adjusted by periodontitis. The AUC calculation was performed in a similar way to that of the protease antibody array analysis.

The level of significance was defined as *p* < 0.05. The statistical analysis was performed using a statistical software, STATA 13^®^, and Stata-Corp (StataCorp LLC, College Station, TX, USA).

## 3. Results

This study included a total of 44 individuals, which were all consenting volunteering adults (≥18 years). Demographics and smoking habits are presented in Table 1.

Firstly, 12 subjects were included to explore the protease human profile in GCF through the protease antibody array analysis: Six healthy controls and six individuals with moderate to severe AD. Both groups presented non-smoking, periodontally-healthy male individuals with the same age (median: 23.5 years, ranging from 20 to 39 years), and an educational level. Regarding the scoring systems for the assessment of AD [13], the results were: A median SCORAD of 49.5 (45.8–85.7 range), a median EASI of 19.95 (16.8–39.6 range), a median BSA of 39.8 (18.5–55.4 range), and for IGA moderate (*n* = 4) and severe (*n* = 2).

In the protease antibody array analysis, 35 human proteases in GCF samples were evaluated. Coordinates of the human protease array membrane are shown in Table 2. The intensity signals of the human protease profiles in the GCF of controls and AD individuals are shown in Figure 1. The levels of 35 human proteases analyzed in the GCF were compared between individuals with AD and healthy controls. Results are presented in Figure 2. The levels of seven proteases, specifically ADAM8 (*p* = 0.0327), ADAM9 (*p* = 0.0094), Cathepsin E (*p* = 0.0282), MMP8 (*p* = 0.0179), Neprilysin/CD10 (*p* = 0.0243), Protein convertase 9 (*p* = 0.0282), and uPA/Urokinase (*p* = 0.0433) were lower in AD patients compared to healthy controls. No inter-group differences were found in the other 28 proteases. The diagnostic ability of ADAM8, ADAM9, Cathepsin E, MMP8, Neprilysin/CD10, Protein convertase 9 (PCSK9), and uPA/Urokinase were further analyzed. Results are presented in Table 3. The biomarkers with the highest diagnostic precision were MMP8 followed by Cathepsin E and ADAM9 (*p* < 0.05). The other 36 proteases were not evaluated since inter-group differences were not statistically significant (*p* < 0.05). A combined model of MMP8, Cathepsin E, and ADAM9 was not considered since it did not converge. Combining MMP8 and ADAMP9, the area under the ROC curve was 90.28%, with a sensitivity and specificity of 83.33%, respectively. However, when comparing the ROC area for MMP8 (0.83) and ADAM9 (0.75), no significant differences were found (*p* = 0.439). Then, the MMP8 ROC area (0.83) was compared to the ROC area obtained from the logistic regression of MMP8 + ADAM9 (0.90), and no significant differences were found (*p* = 0.132). In light of the aforementioned, a single MMP8 use was proposed as a useful protease biomarker rather than the MMP8 + ADAM9 combination, as it would be less expensive and less time consuming, without losing the discrimination and diagnosis accuracy.

Secondly, a more sensitive and more specific analysis was performed through the use of the multiplex bead immunoassay to determine the levels of MMP8 in the GCF in individuals with AD (*n* = 23) and healthy controls (*n* = 21). Based on the multiplex bead immunoassay results, lower levels of MMP8 were observed in the GCF of patients with AD compared to healthy controls (*p* = 0.029), as shown in Figure 3. After adjustment by periodontitis, the significant differences of MMP8 levels between the groups was maintained (*p* = 0.042). The mean GCF concentrations of MMP8 were 64.51 ± 49.46 and 99.26 ± 80.53 ng/mL for AD patients and healthy controls, respectively. Finally, for the MMP8, the ROC area was 0.672 (95% confidence interval: 0.524–0.831), with a cutoff of 82.97 ng/mL, and the sensitivity and specificity were 73.91% and 52.38% for diagnosing AD, respectively (Figure 4).

## 4. Discussion

AD is a chronic inflammatory dermatosis characterized by an abnormal skin-barrier function. The disease is modulated by proteases, and greatly affects the quality of life of individuals who carry it through intractable manifestations and its severity. Although clinical manifestations of AD greatly vary among subjects, at present there are no diagnostic tests that may give a better diagnosis and clinical classification of the moderate/severe AD status.

Emerging evidence has associated several proteases with a defective skin-barrier function in AD [4,5,6,7]. To the best of our knowledge, this is the first study to explore the levels of 35 proteases in the GCF of periodontally-healthy subjects with and without moderate/severe AD without any other diseases and/or habits such as smoking. We chose to measure those 35 proteases using antibody arrays since it facilitates their detection in the same sample, at low volumes, and at the same time. In the present exploratory study, we found lower levels of four catalytic-type proteases in patients with moderate/severe AD: (1) Zinc-binding proteases MMP8, ADAM8, ADAM9, and Neprilysin/CD10, (2) calcium dependent serine-endoprotease Protein convertase 9 (PCSK9), (3) serine-binding protease uPA/Urokinase, and (4) aspartyl-binding protease Cathepsin E; suggesting that these proteases might play an important role in the pathogenesis of AD.

In relation to the possible diagnostic potential of the 35 proteases studied in this research, after logistic model evaluations, it was revealed that a single use of MMP8 allowed for the correct identification of both conditions (AD and healthy subjects), indicating that the GCF could serve as a novel and non-invasive source for potential diagnostic biomarkers of moderate/severe AD. To corroborate this result, MMP8 concentrations in the GCF of individuals with AD and systemically healthy controls were further quantified by a multiplex bead-based immunoassay, considering an adequate sample size. According to the obtained results for the MMP8 concentrations in the GCF of AD patients and systemically-healthy controls (measured by means of an antibody array), we corroborated the presence of lower levels of MMP8 in the GCF of AD patients compared to healthy controls (measured by the multiplex assay). This result was independent of the periodontal status. Therefore, we present for the first time, that the GCF concentrations of MMP8 are downregulated in moderate/severe AD individuals. Notwithstanding, it is widely known that MMP8 is one of the main proteases involved in the destruction of the alveolar bone in periodontitis, and it is a known collagenase that breaks down the type I collagen. Contrary to our results, several studies have reported higher concentrations of this protease in the GCF of patients with periodontitis versus periodontally-healthy controls [14,15]. In fact, it has been specifically reported that the concentration of MMP8 in the GCF of patients with periodontitis versus periodontally-healthy controls was 428.6 (±332.4) versus 95.2 ng/mL (±70.2), respectively [15]. In the present study, similar concentrations of MMP8 were found in the GCF of healthy individuals. As shown in our results, a previous study showed lower concentrations of MMP8 in the saliva from juvenile periodontitis patients versus individuals with gingivitis and healthy controls, suggesting that MMP8 could have a systemic protective and anti-inflammatory role [16]. It is possible that MMP8 presents a dual role depending on its expression. This can be supported in part by an animal study, where knockout MMP8 mice with induced periodontitis showed increased alveolar bone loss in relation to uninfected mice [17]. As a result, our novel findings could imply that MMP8 has a key role, at least to some extent, as a defensive enzyme of the systemic immune response in AD, which is able to perform its protective role remotely in sites distant from the skin lesions.

Regarding the MMP8 detection in skin, it is highly expressed in the course of skin inflammatory processes, conventionally attributed to production/secretion from polymorphonuclear neutrophils (PMN), suggesting that MMP8 has a role in dermic destruction [18]. Moreover, it has been reported that MMP8 is required to develop skin pruritus in the in vitro models [19]. A previous study reported increased levels and activity of MMP8 and MMP9 in skin-wash samples of AD patients versus healthy controls. These authors proposed that MMP8 could play an important role in the pathology of AD, and thus, could be useful as a disease biomarker [7]. Overall, we hypothesize that the lower MMP8 levels observed in the GCF in AD individuals could reflect the high activity and concentrations of MMP8 in the skin of AD patients.

Currently, the AD diagnosis is established on clinical parameters and there are no specific biomarkers available [3,8]. Due to AD being a complex and highly heterogenous disease, it has been reported that it is urgent to discover a reliable diagnostic biomarker to support the clinical diagnostic criteria [8]. MMP8 overexpression is supposed to be the single most helpful biomarker in the diagnosis of periodontitis [20]. In our study, the downregulation of MMP8 showed the potential of this protease as a biomarker to identify individuals with moderate to severe AD. Therefore, the measurement of the concentrations of MMP8 in the GCF of patients by dentists and/or dermatologists could be a beneficial, novel, chair-side diagnostic tool for AD diagnosis since the method is non-invasive and fast (taking approximately 2 min in total for the sample collection), opening the door to future personalized treatments. Therefore, we suggest to further explore the prospective use of MMP8 as a diagnostic biomarker of moderate/severe AD.

Under normal conditions, zinc is involved in the differentiation of keratinocytes and decreases its pro-inflammatory activity. In addition, zinc participates in the expression of filaggrin, one of the key proteins responsible for maintaining the skin barrier function. Previous studies have shown a link between zinc concentrations in overall human fluids and AD. In line with our results, lower levels of zinc in the serum, erythrocytes, and hair have been reported in patients with AD compared to healthy controls. Despite the evident and crucial role of zinc in the maintenance of the skin barrier, little is known about the participation of zinc-binding proteases in the pathogenesis of AD [21]. We believe that it is plausible that the systemic decreased expression of zinc dependent proteases results in the loss of the epidermal barrier function favoring the development of AD. Regarding the other zinc-binding proteases downregulated in the GCF of AD patients, disintegrin and metalloproteinase ADAM8 and 9 act as inducing inflammation or anti-inflammation responses, under specific conditions. ADAM8 and 9 have not been previously studied in AD [22,23]. In line with our results, a previous study reported a higher ADAM9 expression in normal mucosa compared to the inflamed gastric mucosa, suggesting that the decreased ADAM9 may predispose to chronic mucosal inflammation [23]. In a more recent study, the protease profile of saliva samples from oral squamous cell carcinoma (OSCC) patients was studied. Results from this study showed a higher expression of ADAM9 in OSCC patients compared to healthy controls. Specifically, ADAM9 showed 0.45 and 0.767 of the sensitivity and specificity for OSCC, respectively. Since a higher concentration of proteases in saliva varied according to the kind of oral disease, the authors considered that the combination of cathepsin V/kallikrein5/ADAM9 biomarkers was accurate as a diagnostic biomarker of OSCC [24]. We could not find literature regarding the detection of ADAM9 in GCF samples. However, in a previous study, the GCF levels of ADAM8 in patients with periodontitis were decreased post-treatment with a conventional non-surgical periodontal therapy at moderate and severe sites, suggesting that ADAM8 concentrations in the GCF reflect inflammatory and bone-resorbing activities in the periodontal pocket [25]. Another relevant study reported that transgenic mice which have a hypersensitivity reaction and soluble ADAM8 in their circulation, expressed higher E-selectin mRNA levels in inflammatory skin sites compared to non-transgenic mice. This was also true for the expression of L-selectin in PMN from peripheral blood samples. These results suggest that ADAM8 might activate endothelial cells and lead to the upregulation of E-selectin, thus regulating leukocyte infiltration directly or indirectly [22]. Therefore, it could be expected that ADAM8 and 9 would be detectable in the GCF samples of healthy controls, regulating leukocyte infiltration as a defensive mechanism of periodontal tissues [22]. PMN create a barrier that prevent oral bacteria from reaching the connective tissues of the periodontium, thus helping preserve and maintain oral health. When neutrophils are deficient, bacteria prosper causing inflammation, and this has been associated with several diseases such as atherosclerosis, diabetes, and cancer [26,27]. Altogether, our results suggest that ADAM8 and 9 downregulation in the GCF could act as a predispose factor of AD. With respect to Neprilysin/CD10, it has been previously studied in AD. Neprilysin/CD10 is a type 2 cell surface metalloproteinase which protects against excessive skin inflammation by degrading substance P or reducing its levels in the dermal microenvironment [28,29]. A clinical study has also demonstrated higher levels of Neprilysin/CD10 in serum samples of allergy-free versus AD children [30]. This antecedent is in line with our results. Therefore, it is possible that Neprilysin/CD10 plays a key role in downregulating the local inflammation in AD and thus, helps maintain the health of oral tissues.

We also highlight that PCSK9 has a role in the pathogenesis of AD. Our results are supported in part by previous reports on inflammation: PCSK9 levels were decreased in the saliva and serum of chronic periodontitis and rheumatoid arthritis (RA) patients, respectively, compared to healthy controls [31,32]. These results could suggest that PCSK9 may act locally in AD skin. However, the role of PCSK9 in AD requires further investigations.

The urokinase plasminogen activator (uPA) modulates immune-inflammatory and fibrinolytic responses [28,33]. A previous study reported an increased uPA activity in skin samples from acute eczematous AD patients, especially in the deeper layers of the stratum corneum, compared to healthy skin [29]. Similar to our results, the peripheral blood of asthmatics has been shown to have lower levels of uPA compared to healthy controls [34]. However, uPA and suPAR plasma levels did not differ between patients with the atopic eczema/dermatitis syndrome and healthy controls [35]. uPA has been reported to have a central role in regulating the Th1 immune response. An experimental animal study demonstrated that uPA-deficient transgenic mice have an inability to generate IFN-a and IL12 but have increased levels of IL5, a type 2 cytokine. In these mice, the macrophages have an impaired antimicrobial activity and regional lymph nodes contained fewer cells in infected mice [36]. Considered overall, we believe the decreased level of uPA in the GFC in AD patients versus healthy controls may reflect a systemic Th2 response in AD.

Finally, the deficiency of Cathepsin E leads to the spontaneous development of AD-like inflammatory skin lesions in mice, along with the systemic accumulation of IL18 and IL1, a rise in the ratio of CD4+/CD8+ T cells, and the strong polarization of naïve T cells to T helper 2 cells [18]. The mechanism by which the Cathepsin E deficiency is associated with the development of AD could be explained by the fact that IL18 and IL1 are potent inflammatory cytokines. IL18 induces an increase in the serum concentrations of IgE, IL4, and IL13, which are characteristic cytokines of hypersensitivity type 1. In addition, IL1 accelerates the AD-like inflammation initiated by the upregulation of IL18 [37]. In line with our results, a clinical study reported a higher Cathepsin E activity and expression in erythrocyte ghosts of healthy controls versus AD individuals [18], supporting the prospective role of Cathepsin E in the systemic response of AD.

Despite the limitations of the present study including: The use of a very expensive laboratory method for protease screening and selection of molecules with potential diagnostic usefulness as GCF biomarkers of AD, as well as the absence of other proteases commercially-available for multiplex bead immunoassay (or at least, the absence of the ones relevant in this study), we were able to report, for the first time, the GCF protease profiles of moderate/severe AD patients. Our results show that the protease spectrum in both groups presented significant differences in composition in periodontally-healthy subjects. However, it must be noted that this could change in the presence of periodontitis, since the disease usually alters the local composition of the GCF. Moreover, we demonstrated that the GCF concentrations of MMP8 in moderate/severe AD patients is downregulated. These results could be explained by the moderate and severe AD, which causes changes in the circulating/systemic concentrations of these molecules, which in turn impacts the normal protease content in the GCF. However, further studies considering more patients and mild dermatitis individuals should be carried out in the future to further validate these results and the use of these biomarkers for the diagnosis of AD. Likewise, MMP8 levels could be evaluated in the serum of AD patients to increase their diagnostic precision.

To conclude, this study showed that the GCF concentrations of ADAM8, ADAM9, Cathepsin E, MMP8, CD10, Protein convertase 9, and uPA/Urokinase proteases were downregulated in moderate/severe atopic dermatitis patients compared to healthy controls. Based on the multiplex assay results, GCF concentrations of MMP8 were downregulated in AD patients versus healthy controls, suggesting that MMP8 could be a suitable biomarker candidate for a future non-invasive diagnosis of moderate and severe AD.

## Figures and Tables

**Figure 1 biomolecules-10-01600-f001:**
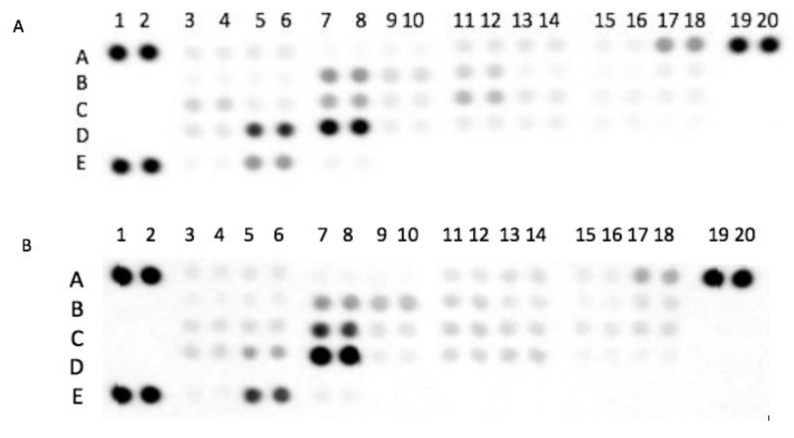
Original membranes of the protease profile in gingival crevicular fluid (GCF) from individuals with (**A**) atopic dermatitis and (**B**) systemically healthy controls (**B**). The following proteases were identified in the membranes: ADAM8 (A3–A4), ADAM9 (A5–A6), Cathepsin E (B3–B4), MMP8 (D5–D6), CD10 (D13–D14), Protein convertase9 (D17–D18), and uPA/Urokinase (E5–E5) with significant differences between both groups. Membranes per group (*n* = 6).

**Figure 2 biomolecules-10-01600-f002:**
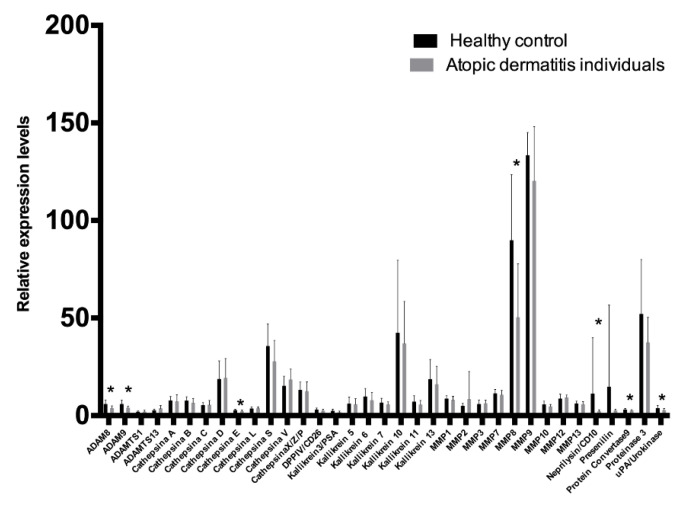
Relative levels of the analyzed proteases in GCF from healthy controls and individuals with atopic dermatitis. Error bars represent the interquartile range. * *p* < 0.05.

**Figure 3 biomolecules-10-01600-f003:**
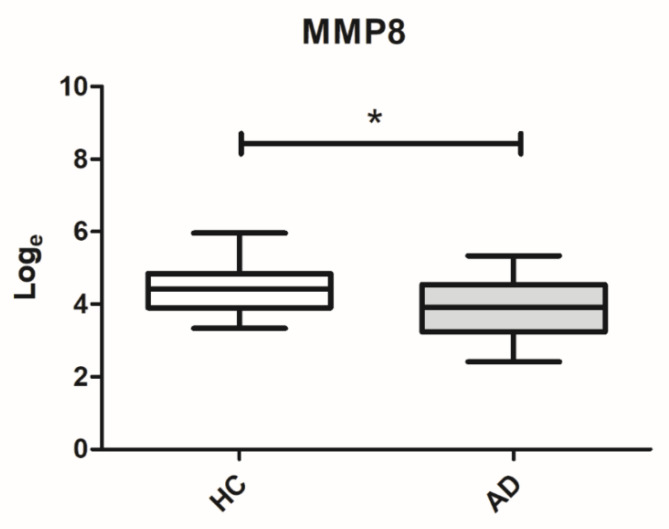
Box plot for MMP8 levels (log scale) in the GCF of individuals with atopic dermatitis and healthy controls. HC: Healthy control; AD: Atopic dermatitis; * *p* < 0.05.

**Figure 4 biomolecules-10-01600-f004:**
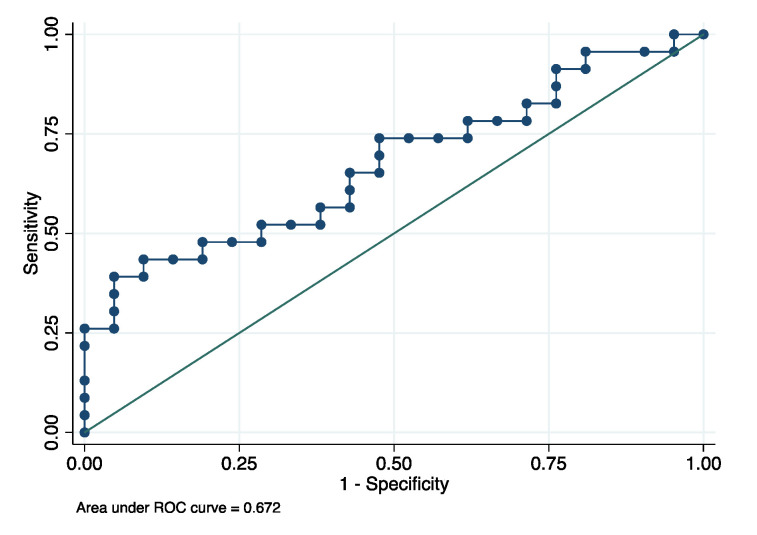
Receiver operating characteristic (ROC) curves for MMP8 to evaluate a biomarker’s ability for moderate/severe AD status.

**Table 1 biomolecules-10-01600-t001:** Demographic data and smoking habits of all study individuals.

Parameters	HC (*n* = 21)	AD (*n* = 23)	*p*
Age (years, mean ± SD)	34.23 ± 12.24	30.52 ± 12.03	0.316
Gender: Male-female (%)	57.14-42.86	56.52-43.48	0.967
Smokers (%)	19.05	4.35	0.176
Periodontitis:			0.529
Mild (%)	23.81	34.78	
Moderate (%)	9.52	17.39	
Severe (%)	0	0	
SCORAD score (mean ± SD)	-	59.36 ± 16.47	
EASI score (mean ± SD)	-	24.73 ± 10.01	
BSA (%)	-	38.14 ± 17.71	

HC: Healthy control; AD: Atopic dermatitis; SCORAD: Scoring Atopic Dermatitis; EASI: Eczema Area and Severity Index; BSA: Body Surface Area. “-” healthy individuals can not present evaluation of psoriasis severity parameters.

**Table 2 biomolecules-10-01600-t002:** Human protease array membrane coordinates.

	1–2	3–4	5–6	7–8	9–10	11–12	13–14	15–16	17–18	19–20
A	RS ^1^	ADAM8	ADAM9	ADAMTS 1	ADAMTS 13	Cathepsin A	Cathepsin B	Cathepsin C	Cathepsin D	RS ^1^
B		Cathepsin E	Cathepsin L	Cathepsin S	Cathepsin V	Cathepsin X/Z/*p*	DPPIV/CD26	Kallikrein 3/PSA	Kallikrein 5	
C		Kallikrein 6	Kallikrein 7	Kallikrein 10	Kallikrein 11	Kallikrein 13	MMP1	MMP2	MMP3	
D		MMP7	MMP8	MMP9	MMP10	MMP12	MMP13	Neprilysin/CD10	Presenilin	
E	RS ^1^	Proprotein convertase 9	Proteinase 3	uPA/Urokinase	Negative Control					

^1^ Reference spot.

**Table 3 biomolecules-10-01600-t003:** Individual diagnostic ability on Odds ratios of biomarkers.

Marker	Threshold	ROC Area (95% CI)	Sensitivity	Specificity	OR (95% CI)	*p*-Value
ADAM8	4.5	0.66 (0.46–0.86)	66.67%	66.67%	4.0 (0.73–21.83)	0.109
ADAM9	4.58	0.75 (0.56–0.93)	75%	75%	9.0 (1.41–57.11)	0.02 *
Cathepsin E	2.62	0.75 (0.56–0.93)	75%	75%	9.0 (1.41–57.11)	0.02 *
MMP8	63.81	0.83 (0.67–0.98)	83%	83%	25.0 (2.92–213.98)	0.003 *
Neprilysin/CD10	2.69	0.66 (0.46–0.86)	66.67%	66.67%	4.0 (0.73–21.83)	0.109
Protein convertase 9	2.84	0.66 (0.46–0.86)	66.67%	66.67%	4.0 (0.73–21.83)	0.109
uPA/Urokinase	3.35	0.66 (0.46–0.86)	66.67%	66.67%	4.0 (0.73–21.83)	0.109

* *p*-value < 0.05.

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
