# Peer review of "Gingival Crevicular Fluid Zinc- and Aspartyl-Binding Protease Profile of Individuals with Moderate/Severe Atopic Dermatitis"

_biomolecules, 2020, doi:10.3390/biom10121600_

Round 1
Reviewer 1 Report
The authors only partly answered my question. The fact that all patients did not have current inflammation is OK for this study, but if they recommend something as a biomarker they should emphasize that current inflammation might affect the results.
Author Response
Following your kind suggestion, we added the following to our discussion: “Our results show that the protease spectrum in both groups presented significant differences in composition in periodontally-healthy subjects, however it must be noticed that this could change in the presence of periodontitis, since the disease usually alters the local composition of the GCF”.
Reviewer 2 Report
I still think that the number of the cases (n=6) is too small, and thus it is very challenging and dangerous to adopt this article for publication.
Author Response
We appreciate your recommendations. We were able to increase the number of case studies in our manuscript from n = 12 (evenly distributed within the atopic dermatitis and control groups) to 23 atopic dermatitis patients and 21 healthy controls. Sample size calculation was made from the results of the multiplex immunoassay of MMP8, resulting in an effect size of 1.1, with a level of significance of 0.05 and power of 0.8. According to this analysis, the minimum sample size required for this study consisted of 15 subjects with atopic dermatitis and 15 healthy controls, hence we were able to successfully surpass the sample size threshold.
Round 2
Reviewer 2 Report
This is an interesting study regarding GCF inspection for AD severity. The authors told that low MMP8 levels in GCF of AD patients reflect high MMP8 levels in the skin. What about serum or plasma MMP8 levels in AD patients vs controls? Is there any study on serum MMP8 levels comparing between AD patients vs controls?
If the difference in serum MMP8 levels between AD and controls is significant, it is more recommended that serum MMP8 inspection is suitable for the diagnosis of AD rather than GCF inspection.
Author Response
REVIEWER
Dear Reviewer,
We appreciate your new recommendation.
In relation of your comments “This is an interesting study regarding GCF inspection for AD severity. The authors told that low MMP8 levels in GCF of AD patients reflect high MMP8 levels in the skin. What about serum or plasma MMP8 levels in AD patients vs controls? Is there any study on serum MMP8 levels comparing between AD patients vs controls?
If the difference in serum MMP8 levels between AD and controls is significant, it is more recommended that serum MMP8 inspection is suitable for the diagnosis of AD rather than GCF inspection.”
A: There is no article available on MMP8 levels in serum or plasma in AD patients. However, we added in the last part of discussion “ Likewise, MMP8 levels could be evaluated in the serum of AD patients to increase their diagnostic precision”.
With Regards,
Dra Alejandra Fernández
This manuscript is a resubmission of an earlier submission. The following is a list of the peer review reports and author responses from that submission.
Round 1
Reviewer 1 Report
Firstly, I would like to congratulate the Authors on their innovative idea to identify biomarkers of AD in GCF. I belive this is a first study to investigate this matter.
The following are the issues with the paper:
- The aim of the study was to investigate protease profiles in GCF. Their evaluation as biomarkers has not been investigated. A simple Odds Ratio analysis is insufficient to determine if a compuond is fit as a biomarker.
- The number of patients for this analysis is very low. Could the authors explain or discuss why they included only 6 patients with AD? As I understand the Authors did a screening with a (probably) very costly method. But that should not exclude the possibility to perform a confirmation of their candidate proteins in a larger cohort using a lot less costly ELISAs.
- As a reader, I was also very curious if the method the authors propose is actually fit for a point-of-care diagnostic. I'd suggest that the authors include in the discussion how much time and effort the GCF diagnostics take in a clinical setting (or skip the assotiation to biomarkers alltogether)
- The conclusions indicate that Molecules indicated in this manuscript "MMP8, ADAM9 and Cathepsin E may be useful as combined diagnostic and therapeutic biomarkers of moderate/severe AD." This conclusion is however not supportet by the presented data. There is no Information about the combined specificity or sensitivity of these markers as diagnostic for AD and severity has not been investigated at all in regard to these markers.
To conclude I think this paper has credible data that should be published but their proposed indication as biomarkers is unbased in the presented data.
Reviewer 2 Report
This is an interesting approach but there are significant concerns. The study is based on the assumption that circulating cytokines and biological molecules from systemic inflammation naturally extravasate into the GCF. This sounds plausible, but it might be expected that their levels are not only influenced by systemic circulation but also by local cytokine milieu e.g. by local inflammation.
In general, the number of subjects is too low (6 patients and 6 controls). Furthermore, no correction for multiple testing has been performed which is essential when a larger number of biomarkers is investigated.
Reviewer 3 Report
This study is too preliminary for publication. The number of cases is too small. Moreover, it is not general to use GCF as the source of biomarkers for AD, a skin disease. For AD patients with periodontitis or gingivitis, clinicians cannot use the proteinases in GCF as a biomarker. Besides the values of proteinases in GCF are changeable by a variety of diseases other than AD. If the authors want this study to be published, I recommend to submit the article to journals specialized in dental and oral diseases.